# Cloud Detection in Remote Sensing Images Based on a Novel Adaptive Feature Aggregation Method

**DOI:** 10.3390/s25041245

**Published:** 2025-02-18

**Authors:** Wanting Zhou, Yan Mo, Qiaofeng Ou, Shaowei Bai

**Affiliations:** 1School of Information Engineering, Nanchang Hangkong University, Nanchang 330063, China; 2304085401015@stu.nchu.edu.cn (W.Z.); ou.qiaofeng@nchu.edu.cn (Q.O.); 2404085404317@stu.nchu.edu.cn (S.B.); 2College of Aeronautics Engineering, Nanjing University of Aeronautics and Astronautics, Nanjing 210016, China

**Keywords:** cloud detection, adaptive feature aggregation, multi-scale, feature fusion

## Abstract

Cloud detection constitutes a pivotal task in remote sensing preprocessing, yet detecting cloud boundaries and identifying thin clouds under complex scenarios remain formidable challenges. In response to this challenge, we designed a network model, named NFCNet. The network comprises three submodules: the Hybrid Convolutional Attention Module (HCAM), the Spatial Pyramid Fusion Attention (SPFA) module, and the Dual-Stream Convolutional Aggregation (DCA) module. The HCAM extracts multi-scale features to enhance global representation while matching channel importance weights to focus on features that are more critical to the detection task. The SPFA module employs a novel adaptive feature aggregation method that simultaneously compensates for detailed information lost in the downsampling process and reinforces critical information in upsampling to achieve more accurate discrimination between cloud and non-cloud pixels. The DCA module integrates high-level features with low-level features to ensure that the network maintains its sensitivity to detailed information. Experimental results using the HRC_WHU, CHLandsat8, and 95-Cloud datasets demonstrate that the proposed algorithm surpasses existing optimal methods, achieving finer segmentation of cloud boundaries and more precise localization of subtle thin clouds.

## 1. Introduction

Remote sensing technology has played an increasingly important role in the past decade, providing technical support for monitoring critical fields such as agriculture [1], forestry [2], oceans [3], soil [4], environmental protection [5], and meteorology [6]. Accurate cloud detection enables the acquisition of detailed cloud data, which facilitate in-depth analysis of weather systems. This has a profound impact on the formulation of disaster prevention and mitigation strategies, the planning of agricultural production, and ensuring aviation transport safety, among other areas. However, cloud cover can obscure ground information, resulting in varying degrees of blurriness or information loss in the collected data, posing significant challenges to research in these critical fields. To enhance the clarity and imaging quality of remote sensing images, accurate cloud detection is imperative. Traditional cloud detection methods can achieve better results under specific conditions but often show greater limitations when faced with complex and variable cloud morphology, changing lighting conditions, and differences in ground cover.

Deep neural networks, endowed with robust feature extraction and representation learning capabilities, possess the ability to accurately identify distinct regions within images and extract objects of interest. As a result, deep learning has become widely popular in numerous application areas. Among them, the convolutional neural network (CNN), as a representative model of deep learning in the field of image processing, is able to automatically learn and extract effective feature representations from a large amount of image data. Johnston et al. [7] applied the CNN to cloud detection, and the algorithm had an amazing ability to apply context information classification, which greatly improved the detection performance. Matsunobu et al. [8] demonstrated that the CNN could directly identify the presence of clouds in remote sensing imagery at a relatively high temporal resolution without the need for auxiliary data. Chai et al. [9] designed a shallow CNN, compared to a deep convolutional neural network, which not only achieved more stable training but also significantly reduced detection costs. Liu et al. [10] proposed a deformable convolutional cloud detection network with an encoder–decoder architecture to address the challenge of CNNs lacking an internal mechanism to handle geometric transformations of clouds. However, the performance of this model may still be constrained when faced with extreme weather conditions or complex surface features.

Attention mechanisms can dynamically adjust the level of attention paid to different regions, and many cloud detection methods use attention mechanisms to improve detection accuracy. Zhang et al. [11] introduced a CNN based on cascaded feature attention and channel attention, enhancing the focus on crucial image features to improve the accuracy of cloud detection. Zhang et al. [12] presented a multi-scale-attention convolutional neural network to improve the accuracy of cloud detection by realizing the acquisition of different receptive fields and the learning of pixel importance through a multi-scale module and an attention module, respectively. Zhang et al. [13] designed integrated color and texture feature attention modules to enhance information extraction and facilitate the web learning of richer details by utilizing dual attention modules. However, this feature extraction method may fail to capture some subtle or complex cloud features, especially when the contrast between clouds and the background is low or when cloud types and morphologies are diverse. Tan et al. [14] proposed an automatic cloud identification method that combines multi-feature fusion with a channel attention mechanism. This method employs MobileNetV2 as the backbone network on the basis of DeepLabV3+ to effectively distinguish between snow and clouds. Nevertheless, the attention mechanisms in these algorithms primarily focus on the extraction of local features, often neglecting global contextual information, which may affect the accuracy and completeness of overall cloud detection.

UNet [15] is a classical network structure in deep learning, which was proposed to solve the problem of biomedical image segmentation. Due to its high scalability and customizability, researchers have improved and optimized it for use in the field of cloud detection according to the needs of specific tasks. Yanan et al. [16] and Guo et al. [17] attempted to incorporate attention mechanisms into symmetrical encoder–decoder network architectures to more accurately differentiate between cloud and non-cloud pixels. Kanu et al. [18] proposed a model based on encoder–decoder architecture and combined Atrous Spatial Pyramid Pooling and separable convolution to optimize the network and improve the accuracy of cloud detection. Hu et al. [19] presented Cloud Detection UNet, an encoder–decoder network architecture specifically designed for cloud detection in remote sensing images, which is capable of delicately segmenting cloud edges and precisely identifying thin clouds against complex ground backgrounds. Yin et al. [20] employed Resnet 50 instead of the UNet3+ feature extraction network and added an attention module to extract deep features for the accurate detection of clouds and snow. Li et al. [21] aimed to embed a dense global context block into the UNet framework for thin cloud detection. Li et al. [22] enhanced the information flow by cascading two U-shaped networks and utilizing residual connections, and their improved hopping connections facilitate multi-scale feature exploitation, which aids in the identification of thin clouds. Yet, along with the performance improvement, this algorithm also results in an increase in parameter volume and computational complexity, posing key challenges that urgently need addressing in practical applications.

Originally designed for natural language processing tasks, Transformer’s powerful sequence processing capabilities and flexibility have led to corresponding extended applications in cloud inspection. Wen et al. [23] proposed an improved cloud detection model that integrates Transformer with a multi-scale feature extraction module to efficiently capture global and local multi-scale features in cloud regions. Ma et al. [24] combined the advantages of both Transformer and the CNN to design a hybrid CNN–Transformer network with differential feature enhancement to enhance finer detail extraction and establish long-range dependencies. Singh et al. [25] presented a spatial–spectral attention transformer network that introduces a spatial–spectral attention module at its core, forgoing traditional convolutional methods and directly utilizing image patches to generate enhanced feature maps, thereby enhancing cloud detection performance. Tan et al. [26] proposed an innovative Transformer-based cloud detection method based on Transformer, which optimizes cloud detection by introducing Cyclic Refinement Architecture to enhance the resolution and quality of feature extraction and to effectively retain key details.

Cloud shadows are areas of darkness that appear on the Earth’s surface when sunlight is obstructed by clouds. The presence of these shadowed areas can interfere with the acquisition of surface information. Therefore, a large number of methods for cloud and cloud shadow detection have also been investigated in recent years. Wieland et al. [27] designed a CNN based on UNet improvement to segment clouds and cloud shadows in remote sensing images. Qu et al. [28] established a strip pooling channel spatial attention network to accurately extract the local position information of clouds and their shadows and improve the accuracy of edge segmentation. Mohajerani et al. [29] used a new filtered Jaccard loss based on Cloud-Net [30] for training and developed a sunlight direction-aware data augmentation technique for cloud shadow detection. Li et al. [31] proposed a novel Transformer-based cloud shadow detection algorithm, which uses a hierarchical Transformer structure in the encoder to extract cloud shadow features and a multilayer perceptron in the decoder for the fusion of features and pixel classification.

Although deep learning has shown great potential for application in the field of cloud detection in remote sensing imagery, many existing methods are still not robust enough to deal with the fine definition of cloud boundaries and the accurate identification of thin clouds. Specifically, thick clouds in remote sensing images often occupy prominent positions in stacked patterns, contrasting sharply with background pixels, and are relatively intuitive to detect; however, cloud boundaries are intricate and complex, and accurately depicting complete cloud boundaries has become a major challenge. What is even more difficult is that thin cloud regions scattered in images are difficult to distinguish from the background pixels, which undoubtedly increases the detection difficulty. Therefore, how to perfectly outline the cloud boundaries and accurately capture thin cloud regions has become a key challenge to be solved. To address this challenge, we propose the NFCNet algorithm model based on coding and decoding structures. The innovative contributions of this study are summarized as follows:The HCAM possesses the capability to capture multi-scale information, offering a more extensive understanding of the intricate morphology and hierarchical organization of clouds. Furthermore, it expertly assigns differentiated weights to feature channels, thereby enhancing the crucial feature details essential for cloud detection.The SPFA module employs a novel adaptive feature aggregation method, which not only comprehensively reveals the spatial distribution characteristics of cloud layers but also precisely focuses on and meticulously captures local details, transcending the limitations of merely acquiring local features. Consequently, this method excels in achieving both the precise identification of delicate thin clouds and intricate delineation of cloud boundaries simultaneously.The use of the DCA module fuses the hierarchical details of encoding and decoding to guide the detail recovery of the feature maps, enabling more accurate localization of clouds and finer outlining of cloud boundaries.In the encoding stage, the model skillfully fuses high-resolution and low-resolution information to achieve the complementary enhancement of fine details and extensive contextual information. This complementary information enables it to flexibly adapt to diverse application scenarios and improve its recognition of thin clouds.

The layout of the subsequent sections is as follows. Section 2.1 describes our proposed NFCNet in detail. The dataset used and the validity of our algorithm through extensive experiments are presented in Section 3. Finally, we give conclusions in Section 4.

## 2. Methodology

In this section, we will detail the proposed deep learning cloud detection network architecture based on deep learning. Firstly, we will introduce the overall framework of the network. Subsequently, we will elaborate on the three pivotal submodules that constitute the network. Lastly, we will introduce the network training loss function employed.

### 2.1. Overview

As depicted in Figure 1, the network architecture of the proposed model is improved by the use of encoders, decoders, and hopping connection operations designed to achieve efficient differentiation between clouds and background. The cloud map undergoes four levels of encoder operations to gradually extract and compress the features of the input image to form a coded feature map. In the decoding stage, the feature map is upsampled level by level to recover the spatial dimension of the image to match the input feature map size.

The overall model consists of the HCAM, SPFA module, and DCA module, which can effectively capture semantic information and detailed features in an image and cope with the challenges of the complex shapes of cloud maps and highly similar regions to the background. In the encoding stage, the HCAM is used to extract different scale features in the image, which cover all aspects from local details to global structure, and pays special attention to the critical detailed information in the cloud image, thus greatly improving the detection of small-scale objects. Meanwhile, the SPFA module is introduced to enhance the feature representation in the channel and spatial dimensions, effectively compensating for the spatial details lost due to the downsampling operation. In the decoding stage, the SPFA module further enhances the key information in the upsampled feature map to achieve the optimization of the segmentation effect. The high-resolution feature maps generated by the encoder are combined with the same-resolution feature maps obtained after the upsampling process to achieve complementary information, and this combination constitutes one end of the input to the DCA module. The other input is derived from the output of the decoder after upsampling. The DCA module skillfully integrates the shallow details from the encoder with the deep semantic features of the decoder, aiming to balance the semantic and detailed information, enhance the richness of the cloud map features in the decoding stage, and then improve the efficiency and feasibility of decoding.

### 2.2. HCAM

In deep neural networks, the receptive field is the range of influence of the output neurons of a layer in the neural network on the input data, and the magnitude of its value indicates the amount of contextual information that can be accessed and determines the extent to which the neural network understands the input data, which has a direct impact on the accuracy of the target recognition.

To improve the network’s ability to perceive local features and capture more contextual information, it is necessary to increase the receptive field. Nevertheless, the study of Zhou et al. [32] showed that the receptive field of convolution is limited; for deeper networks, the perceptual domain is much smaller than the theoretical one, making it difficult to capture long-range feature correlations. The receptive field can be expanded layer by layer by increasing the number of convolutional layers, but Sun et al. [33] showed that for DNNs with a limited number of hidden units, increasing the depth is not always good, and too deep a network may lead to problems such as gradient vanishing or gradient explosion. He et al. [34] proposed Spatial Pyramid Pooling; although it can extend the receptive fields by pooling operations at different scales, the sizes of these receptive fields are fixed and cannot fully adapt to changes in the sizes and shapes of all targets in the input image. In addition, the parallel use of multiple pooling of different sizes to fuse multi-scale features inevitably produces some degree of information loss. To overcome these limitations, we propose the use of dilated convolutions with varying dilation rates [35] to increase the receptive field, thereby enhancing the semantic information of deep features. This is beneficial for improving the detection performance of small cloud targets. The advantage of dilated convolution lies in preserving the original network’s receptive field while not losing spatial resolution. This feature enables it to effectively address the issue of reduced spatial resolution associated with the subsampling process, and maintaining spatial resolution is crucial for cloud detection tasks involving multi-detail scenes.

Each channel of the feature map carries different feature information, and the degree of contribution of these channels in cloud detection tasks varies as well. The SE Block [36] dynamically allocates feature weights based on the learning loss to match higher weights for key feature channels and reduce the interference of irrelevant regions on the prediction results. Firstly, the features are compressed spatially through global average pooling, transforming each two-dimensional feature channel into a real number with a global receptive field. Secondly, after obtaining the global information, the generation and selection of feature channel weights are achieved in a parameterized manner. Finally, the adjusted importance weights are multiplied element-wise with the original feature map, effectively recalibrating the original features along the channel dimension. This further enhances the expressive power and specificity of the features.

Combining these two points, we propose the HCAM, as shown in Figure 2. Firstly, a 3 × 3 convolution with a stride of 1 is used to reduce the number of channels, thereby not only alleviating the computational burden but also adeptly capturing local features. Secondly, dilated convolutions with different resolutions are employed to extract features at various scales, with resolutions set to 1, 6, 12, and 18, respectively. Thirdly, a 1 × 1 convolution is utilized to adjust the dimensions of the multi-scale features fused in the previous step. Fourthly, the SE Block mechanism is integrated to dynamically assign importance weights to the channels, focusing on the key features in the cloud detection task. Fifthly, the calibrated channel importance weights are multiplied element-wise with the output feature map from step three to achieve feature reweighting. Finally, a 3 × 3 convolution is applied to obtain the refined feature map output, where the SE Block part and the final output feature maps are given by the following equations:(1)$$\begin{array}{c} S_{c}=GAvgPool\left(F_{out}\right)=\frac{1}{H\times W}\sum_{i=1}^{H}\sum_{j=1}^{W}F_{out}\left(i,j\right) \end{array}$$(2)$$\begin{array}{c} Z_{out}=\delta \left[W_{2}\sigma \left(W_{1}S_{c}\right)\right],\;\;W_{1}\in R^{\frac{c}{r}\times r},W_{2}\in R^{r\times \frac{c}{r}} \end{array}$$(3)$$\begin{array}{c} Out=\sigma \left(BN\left\{{Conv}_{3\times 3}\left[F+Softmax\left(F_{out}S_{out}\right)\right]\right\}\right) \end{array}$$
where $F_{out}$ is the output feature map after 1 × 1 convolution for dimension adjustment, *F* denotes the local feature map extracted by the first 3 × 3 convolution, GAvgPool is the global average pooling, BN is BatchNormal, σ is the Relu activation function, δ is the sigmoid function, *H* and *W* denote the height and width of the feature maps, and $W_{1}$ and $W_{2}$ refer to the two fully connected operations, where the first fully connected layer achieves feature dimensionality reduction and fusion by decreasing the number of channels, while the second fully connected layer recalibrates the reduced-dimensionality features.

### 2.3. SPFA

When an image undergoes downsampling, it loses contextual information and details, which makes it difficult to recover the lost boundaries and texture-rich region features during the upsampling process to restore the image resolution. Therefore, it is particularly important to enhance the network’s ability to capture long-range semantic relationships to achieve more accurate boundary detail predictions. Guo et al. [37] used multiple strip convolutions in a pyramid structure to obtain multi-scale global contextual information, enhancing the detection of strip-shaped targets. Liao et al. [38] used a parallel strip convolution enhancement network to improve the perception ability of the target area. To enhance the semantic information of deep features, we use grouped strip convolution [39] to extract multi-scale features. Strip convolution increases the model’s receptive field in that dimensional direction by dividing the regular convolution into long strips of convolution kernels in the horizontal and vertical directions, allowing the model to capture long-distance spatial dependencies more efficiently.

In order to enhance the model’s attention to detailed features during downsampling and at the same time realize the effective recovery of key features in the upsampling stage, we not only need to pay attention to the extraction of channel features but also need to strengthen the capture and utilization of spatial information. The SE Block used in the HCAM can dynamically assign channel importance weights but lacks access to spatial information, whereas the cloud detection task requires not only identifying the presence of clouds but also accurately locating their positions. Inspired by the CBAM’s [40] ability to focus on both channel and spatial information, several excellent algorithms improve the use of the CBAM to improve target detection accuracy. For example, Cheng et al. [41] designed multi-scale fused attention modules for more accurate depth prediction. Meng et al. [42] proposed a multifunctional fused attention gate module to effectively suppress irrelevant background interference. The channel attention module relies on pooling operations when extracting channel features, but this process inevitably leads to the loss of some information [34]. Furthermore, due to the limitation of the receptive field, the module has constraints in capturing global information, making it difficult to fully obtain the comprehensive features of the entire input data. To overcome this limitation, the self-attention [43] mechanism is introduced. Self-attention improves the detection of fine thin clouds in cloud maps by capturing the correlation between two elements to obtain dense pixel-level contextual information. Specifically, this method calculates the similarity between Query and Key and then uses these similarities as weights to perform a weighted summation on Value. This allows for the effective capture and fusion of important information within the input sequence, thereby generating a comprehensive representation that contains both the original information and contextual information.

Combining the above analyses, we introduce the SPFA module, which aims to efficiently compensate for the loss of information details in the downsampling stage and significantly enhance detail expressiveness during the upsampling process. The detailed structure of the SPFA module is shown in Figure 3. First, the framework incorporates a parallel multi-scale stripe convolution mechanism to accurately extract and fuse multi-level features. The expression for this process is as follows:(4)$$\begin{array}{c} e_{i}=\left\{\begin{array}{c} I_{n},\;\;i=0,I_{n}\in R^{C\times H\times W}\\ GAP\left(I_{n}\right),\;\;i=1\\ Conv_{p\times q}\left(I_{n}\right),\;\;i=2∼7 \end{array}\right. \end{array}$$(5)$$\begin{array}{c} F_{i}=Conv_{1\times 1}\left[Cat\left(e_{i},e_{i+1}\right)\right],\;\;i=2,4,6 \end{array}$$(6)$$\begin{array}{c} X=Cat[Conv_{1\times 1}\left(e_{1}\right),F_{2},F_{4},F_{6}] \end{array}$$
where $I_{n}$ represents the input feature map, with stripe convolution groups corresponding to 2, 4, and 8, respectively, from smaller to larger scales.

Second, to broaden the model’s global perspective, we cleverly integrate a self-attention mechanism to apply global attention weighting to the feature maps while simultaneously combining a spatial attention module to achieve the precise localization of cloud pixel positions. Additionally, we deploy an MLP in parallel, with a dimensionality reduction rate of 16 set through experimental testing. The first layer of the MLP reduces the input feature dimension to 1/16 of its original size, while the second layer is responsible for restoring it to the original dimension. Subsequently, we perform a meticulous pixel-level concatenation of the output feature maps from the MLP with the feature maps processed by the attention mechanism to further enrich the feature representation, where the mathematical equation of the detailed process of self-attention combined with spatial attention can be expressed as follows:(7)$$\begin{array}{c} Q,K,V={Conv}_{3\times 3}\left(V\right) \end{array}$$(8)$$\begin{array}{c} Att=X\times {Conv}_{1\times 1}\left[Softmax\left(\frac{QK^{T}}{\sqrt{dk}}\right)V\right],\;\;X\in R^{4C\times H\times W} \end{array}$$(9)$$\begin{array}{c} F_{s}avg=AvgPool\left(Att\right),\;\;F_{s}max=NaxPool\left(Att\right) \end{array}$$(10)$$\begin{array}{c} F_{s}=Att\times Softmax\left\{{Conv}_{7\times 7}\left[Cat\left(F_{s}avg,F_{s}max\right)\right]\right\} \end{array}$$(11)$$\begin{array}{c} Out=Conv_{1\times 1}\left(MLP\times X+F_{s}\right) \end{array}$$(12)$$\begin{array}{c} Out\in R^{C\times H\times W} \end{array}$$

### 2.4. DCA

The encoder part goes through the HCAM, SPFA module, and downsampling operations to gradually extract high-level semantic features of an image. Still, at the same time, the spatial resolution is gradually reduced, causing the decoder to reconstruct the target data without sufficient spatial location information and local details to affect the generation of a high-quality output. Therefore, the introduction of the DCA module realizes the effective integration of shallow and deep feature maps, which play a key role in the process of information feature fusion at different scales and semantic levels.

The DCA module is designed as an efficient feature fusion mechanism. It first performs multi-scale modeling on the input features at the decoding end to obtain multi-scale features that contain rich semantic information to enhance the detection performance of cloud boundaries with different morphologies. Then, the DCA module skillfully fuses the features at the encoding end with the multi-scale features at the decoding end to realize feature complementation and enhancement. Subsequently, the fused features are injected into the decoding phase, ensuring that during the gradual restoration of the image’s spatial resolution, the image details can be effectively and finely recovered, thereby enabling more accurate segmentation decisions to be made.

The proposed skip connection DCA module is shown in Figure 4; input1 is the shallow feature map and input2 is the deep feature map. First, we let the shallow features reduce the spatial dimensions by global average pooling, with the channel dimensions still retained. Since the channel averages of the global average pooling output features directly reflect the confidence level of the corresponding category of the channel, a 1 × 1 convolution is considered to learn the weights between different channels to adjust the importance of the features more flexibly. The mathematical calculation equation for the process is as follows:(13)$$\begin{array}{c} F_{Lout}=\sigma \left(BN\left\{{Conv}_{1\times 1}\left[GAvgPool(Input1)\right]\right\}\right) \end{array}$$

Next, the output of the decoder is used as an input to the input2 port of the DCA module in the following steps. In step 1, the deep features are convolved 1 × 1 to obtain $F_{D}$, reducing the number of channels in the feature map. In step 2, the feature map $F_{D}$ is executed as strip convolution operations with four different-sized convolution kernels to capture different scale features and increase the diversity of feature representations, after which the four outputs are combined in channel dimensions to obtain $F_{D1}$. In the third step, a 3 × 3 convolution operation is performed on the output feature $F_{D1}$ for further feature extraction and smoothing process. The equation is expressed as follows:(14)$$\begin{array}{c} F_{D}=\sigma \left\{BN\left[{Conv}_{1\times 1}\left(Input2\right)\right]\right\} \end{array}$$(15)$$\begin{array}{c} F_{D1}=Cat[{Conv}_{1\times 3}\left(F_{D}\right),{Conv}_{3\times 1}\left(F_{D}\right),\\ {Conv}_{1\times 5}\left(F_{D}\right),{Conv}_{5\times 1}\left(F_{D}\right)] \end{array}$$(16)$$\begin{array}{c} F_{Dout}={Conv}_{3\times 3}\left(F_{D1}\right) \end{array}$$

Third, $F_{Dout}$ and $F_{Lout}$ are multiplied element by element, and a 1 × 1 convolution outputs the shallow and deep fused feature maps.(17)$$\begin{array}{c} F_{o}=\sigma \left(BN\left\{{Conv}_{1\times 1}\left[F_{Dout}\times F_{Lout}\right]\right\}\right) \end{array}$$

### 2.5. Loss Function

In deep learning image segmentation, the output prediction results hinge on the design of the network architecture and the choice of the loss function, which is a pivotal component in the model optimization process and defines how the difference is measured between the model prediction and the true label. Different loss functions will steer the model towards learning in distinct manners, thereby influencing the performance and behavior of the model. By taking into account the types of datasets and the characteristics of cloud pixels, we chose the cross-entropy loss function to train the network model and employed the Adam optimizer to optimize the loss function. The cross-entropy loss quantified the discrepancy between the output probability distribution of the model and the probability distribution of the real labels, effectively distinguishing the difficulty of samples during the training process and facilitating faster convergence. The mathematical expression of the cross-entropy loss function is defined as follows:(18)$$\begin{array}{c} L=-\frac{1}{N}\sum_{i\in N}\left\{p\left(x_{i}\right)log\left[q\left(x_{i}\right)\right]+\left[1-p\left(x_{i}\right)\right]log\left[1-q\left(x_{i}\right)\right]\right\} \end{array}$$
where *N* denotes the number of pixels in each image, $p(x_{i})$ denotes the true value of the cloud label, and $q(x_{i})$ is the prediction result.

## 3. Experimental Results

### 3.1. Dataset

The CHLandsat8 dataset is designed for cloud detection in high-resolution remote sensing images of Chinese regions. The dataset contains 64 full-scene images from different regions, covering diverse weather conditions, land cover types, and cloud morphology. In the experiments, the original 8000 × 8000 pixel images were cropped to 352 × 352 pixel blocks to reduce the computational complexity; finally, 22,616 images were obtained for training and 10,080 images were obtained for testing and validation. During the training process, we resized the images to 256 × 256 and randomized and flipped the images to increase the diversity of the data.The HRC_WHU dataset [44] is a high-resolution cloud cover validation dataset established by Wuhan University. It comprises 150 high-quality images with resolutions ranging from 0.5 to 15 meters, comprehensively covering five major land surface types: water, vegetation, urban areas, snow and ice, and barren land. The images possess a pixel size of 1280 × 720. During preprocessing, each image was randomly cropped into 80 images of 256 × 256 pixels, with blurry and duplicate images being eliminated. Consequently, a total of 8800 images were utilized for training, while 3150 images were designated for testing and validation. Throughout the training process, the images underwent random permutations and flips.The 95-Cloud dataset [29] represents an expansion of the 38-Cloud dataset [30], which inherits the characteristics of the 38-Cloud dataset and adds more data and samples. This expanded dataset aims to support more sophisticated cloud detection and segmentation tasks. The training patches for the dataset were extracted from 75 Landsat 8 Collection 1 Level-1 scenes. Given the presence of black areas around the Landsat 8 image scenes, empty pixels occupying over 80% of the training patches were excluded to enhance training efficiency. In the training process of the model, we merged the three RGB channels of the dataset to form a three-channel data image and resized the original pixel size of the training images from 384 × 384 to 256 × 256, applied random permutations, and flipped the images. To further improve the model’s generalization capability, the 38-Cloud and 95-Cloud datasets were combined, with 80% of the combined data used for training and the remaining 20% reserved for testing and validation.

### 3.2. Evaluation Metrics

We assessed the accuracy and reliability of the segmentation results from multiple dimensions by employing some common evaluation metrics commonly used in cloud detection tasks.(19)$$\begin{array}{c} Accuracy=\frac{TP+TN}{TP+TN+FP+FN} \end{array}$$(20)$$\begin{array}{c} Dice=\frac{2\times TP}{2\times TP+FP+FN} \end{array}$$(21)$$\begin{array}{c} mIoU=\frac{TP}{TP+FP+FN} \end{array}$$(22)$$\begin{array}{c} Precision=\frac{TP}{TP+FP} \end{array}$$(23)$$\begin{array}{c} Recall=\frac{TP}{TP+FN} \end{array}$$(24)$$\begin{array}{cc} F1 & =\frac{2\times Precision\times Recall}{Precision+Recall} \end{array}$$

In the above equation, TP is True Positive, TN is True Negative, FP is False Positive, and FN is False Negative.

### 3.3. Ablation Experiment

In this section, in order to evaluate the contribution of each module to the overall model, we detail how we conducted per-module ablation experiments on the HRC_WHU dataset. Among other things, to present the advantages of the HCAM, we chose to alternate the basic two-time 3 × 3 convolution instead of using the HCAM as a baseline on the codec.

As shown in the above Table 1, compared to the baseline model, the introduction of the HCAM achieved a significant improvement in model performance, showing an all-round optimization effect. Specifically, Dice, IoU, Re, and F1 were improved by 2.44%, 4.03%, 2.54%, and 2.39%, respectively, presenting the advantages of HCAM’s ability to capture key features and especially perform well in enhancing the completeness of the model’s ability to recognize all relevant instances. The SPFA module was further fused to realize the deep mining of multi-scale contextual information and effective compensation of detailed features, which significantly enhanced the detection accuracy of the model. It is worth noting that when the output of the SPFA module was directly used as the input of the DCA module; the comprehensive evaluation of Pre and Re was degraded, whereas combination with high resolution through upsampling as the input of the DCA module could produce substantial improvements in the model. The superiority of the fusion of high-resolution and low-resolution strategies at the coding end of the DCA module could be demonstrated, providing strong support for achieving finer detail recovery in the decoding stage.

### 3.4. Comparison Experiment

To verify the performance of our algorithm on cloud detection, we compared the proposed model with other excellent cloud detection models on three datasets: HRC_WHU, CHLandsat8, and 95-Cloud. Specifically, we compared it to UNet [15], Deeplabv3+ [45], SAtt-UNet [16], Cloud-AttU [17], DCNet [10], CRSNet [46], BABFNet [47], MCDNet [48], and AFMUNet [49]. Among them, UNet and Deeplabv3+ are classical network models in semantic segmentation, which have achieved remarkable results in the field of cloud detection. Since our algorithm is based on an encoder–decoder architecture, we chose the SAtt-UNet, Cloud-AttU, and DCNet models for comparison. These three models, along with CRSNet, BABFNet, MCDNet, and AFMUNet, are all specifically designed for cloud detection tasks and have achieved high accuracy.

#### 3.4.1. Cloud Detection Results on HRC_WHU Dataset

Table 2 shows the results of the experimental data metrics’ output on the HRC_WHU dataset for the proposed algorithm and the nine compared models. It is clearly seen that our proposed algorithm was optimal in the five metrics of PA, Dice, IoU, Pre, and F1, and although Re was not the highest, there was only a small difference compared to the best-performing models. The excellent performance of these metrics reflects the advantages of our proposed algorithm in terms of comprehensive performance, especially in the handling of complex backgrounds and diverse cloud shapes, which shows high stability and adaptability.

The HRC_WHU dataset contained scenarios with five different land cover types. Among them, the cloud detection task with snow/ice and water as the background was particularly challenging and highly susceptible to false detection and missed detection problems. In order to further validate the performance of the NFCCNet algorithm in different scenarios, we selected six models that performed well in the overall dataset and evaluated them in exhaustive experiments. As shown in Figure 5, our algorithm exhibited optimal performance under snow/ice and water backgrounds, almost comprehensively outperforming other algorithms. Meanwhile, NFCNet also presented the highest ACC and IoU in all other scenarios, which fully proves that our algorithm was able to show better stability and anti-interference ability when facing highly disturbed backgrounds.

To more intuitively demonstrate the improvement in the proposed algorithm in cloud boundary identification and thin cloud detection capabilities, we screened six sets of images from the test set for the comparative demonstration of model prediction effects. Figure 6 displays the segmentation results of various algorithms on six sets of images, with red marks indicating missed detection pixels, green marks indicating falsely detected pixels, white representing cloud areas, and black representing the background. From the results, it is easy to see that all the algorithms showed high accuracy in localizing thick cloud regions, but there were different degrees of leakage and misdetection when dealing with cloud boundaries and thin cloud regions. Among them, Cloud-AttU, DCNet, and AFMUNet were unable to detect the fine thin cloud pixels in the third set of images, and although other comparison algorithms made some improvements in this regard, there were many falsely detected pixels for the cloud boundary region. For the background area where cloud boundaries and thin clouds intertwined in the second set of images, most of the comparison algorithms almost predicted this area as cloud. And, for the thin cloud boundary regions in the fourth and fifth set of images, all the algorithms had a hard time outlining them completely. In contrast, our proposed algorithm performed optimally in depicting cloud boundaries and detecting thin cloud areas, with the minimum number of falsely detected and missed pixels. Furthermore, for the last set of images, our algorithm could still detect the cloud outline well even in the presence of a highly disturbed background, demonstrating its superior anti-interference capability. This outstanding performance is attributed to the innovative design of our algorithm: by utilizing dilated convolutions with different dilation rates to extract features, the receptive field was effectively expanded, enhancing the detection capability for fine and thin clouds. Additionally, we introduced the channel attention mechanism, which enabled the algorithm to focus on the most critical information by differentially weighting the features of each channel, thus further enhancing the algorithm’s ability to discriminate clouds in complex backgrounds.

#### 3.4.2. Cloud Detection Results on CHLandsat8 Dataset

Table 3 displays the experimental data indicator results outputted by all algorithms on the CHLandsat8 dataset. From the data analysis, it can be observed that our algorithm exhibited superior performance compared to other algorithms in terms of the indicators PA, Dice, IoU, and F1. Although the Pre and Re data did not attain the highest values, the comprehensive evaluation indicator F1 achieved the best result, which effectively demonstrates that our model possessed good stability and robustness.

To more intuitively demonstrate the performance advantages of our algorithm, we also selected six sets of images from the test set for the predictive comparison of the models. These images were chosen with a focus on scenarios where cloud boundaries were intricate and the detection of thin cloud areas was particularly challenging. As shown in Figure 7, in the first, fourth, and fifth sets of images, the comparison algorithms suffered from severely missed detections when capturing thin cloud regions that were highly similar to the background, indicating a lack of sufficient context extraction capability. In the second set of images, detecting thin cloud boundaries was extremely challenging, and apart from BABFNet, the other algorithms handled the details roughly, failing to finely depict the boundaries of the thin clouds. In the third and sixth sets of images, the comparison algorithms performed unsatisfactorily in detecting small cloud targets, mainly due to the insufficient processing of spatial location information, making precise localization difficult. In contrast, our proposed algorithm demonstrated significant advantages in thin cloud localization and boundary detection. This was primarily attributed to the use of parallel multi-scale stripe convolution technology in our algorithm, which significantly enhanced the network’s ability to capture long-distance semantic relationships, thereby improving the detection accuracy for thin clouds. Furthermore, we cleverly combined the self-attention mechanism and spatial attention mechanism to precisely guide the model in compensating for boundary details, effectively mitigating the significant loss of detailed information caused by the downsampling process. Going further, the fusion of feature information by the DCA module was injected into the decoding stage, which provided strong support for detail recovery and enabled the model to depict the cloud boundary morphology in a more delicate and precise manner.

It is noteworthy that a major challenge our algorithm faced in the cloud detection task of this dataset was the high number of falsely detected pixels. The root of this issue lay in certain defects in the design of the feature extraction algorithm. These deficiencies resulted in the algorithm mistakenly classifying more non-cloud areas as clouds during the recognition process, thereby affecting the overall detection accuracy.

#### 3.4.3. Cloud Detection Results on 95-Cloud Dataset

To more comprehensively validate the performance of the proposed method in cloud detection, we also conducted comparative experiments on the 95-Cloud dataset. Table 4 presents the experimental data and performance metrics of various algorithms on the 95-Cloud dataset. As can be seen from the data in the table, our algorithm outperformed the comparison model in all evaluation metrics except for the Re data. Specifically, the BABFNet model achieved the best performance on the comparison algorithm, while our algorithm was improved by 0.51%, 1.38%, 1.83%, 1.09%, 1.58%, and 1.42% in PA, Dice, IoU, Pre, Re, and F1, respectively. This fully demonstrates the feasibility of the innovative model functions of our algorithm, as well as the superiority of its integration strategy, enabling more accurate cloud detection.

We also selected six sets of images from the test set for the visual comparison of the cloud detection performance of each model. These images all met the criteria of having complex cloud boundaries and difficult-to-identify, thin cloud areas. As shown in Figure 8, from an overall perspective, the segmentation results of UNet and DeeplabV3+ were relatively rough, with a large number of false detections. Especially in the first three sets of images, these two algorithms tended to mistakenly classify background areas with similar textures to clouds as clouds. In comparison, CRSNet, BABFNet, and AFMUNet exhibited relatively better detection performance, demonstrating stronger anti-interference capabilities when processing background pixels. However, they still suffered from a certain degree of false detections and missed detections. In the middle background area of the fourth set of images, there was high similarity between the background’s pixels and those of thin clouds. Due to the lack of effective feature differentiation and detail capture capabilities, UNet, DeeplabV3+, CRSNet, and DCNet mistakenly identified a large number of background pixels as clouds when processing this area. The last two sets of images similarly highlighted the deficiencies of the comparison algorithms in localizing thin cloud regions.

Our proposed algorithm achieved certain results in reducing false detections and missed detections, particularly in depicting fine thin cloud regions and boundary details with higher precision. This was attributed to our use of parallel dilated convolutions to capture multi-scale information, enabling the model to better understand the morphology and distribution of clouds. Meanwhile, the use of the attention mechanism effectively reduced the loss of detailed information and realized the complementary enhancement of fine details and extensive contextual information by efficiently fusing the information features of different scales and semantic levels in the process of recovering the image resolution, which further improved the quality of recovering the boundary pixels.

However, our algorithm also had deficiencies in restoring the details of thin cloud boundaries. Specifically, the boundaries of the thin cloud regions were often not clear enough in the detection process, and there was a loss of some detailed information, which affected the accuracy and credibility of the cloud detection results to a certain extent.

### 3.5. Experimental Analysis and Prospects

In this study, we conducted comprehensive cloud detection experiments on three representative remote sensing image datasets. The experimental results indicate that our algorithm performed excellently across all three datasets in terms of comprehensive evaluation metrics, outperforming other advanced algorithms for comparison. Notably, the CHLandsat8 dataset, due to its wide range of geographical and climatic conditions, significantly increased the complexity of the cloud detection task, leading to the inferior performance of various algorithms on this dataset compared to the other two. In contrast, although the 95-cloud dataset also exhibited diversity, its data distribution was relatively more concentrated, which to some extent reduced the detection difficulty. Meanwhile, the use of the HRC_WHU dataset focused more on testing the models’ ability to accurately capture detailed information, which was not the primary challenge posed by the CHLandsat8 dataset.

By visualizing and analyzing the results of our experiments, we drew the conclusion that the proposed algorithm demonstrated superior performance on the HRC_WHU dataset. Specifically, it exhibited a low count of falsely detected and missed detected pixels while possessing notable advantages in capturing boundary details and identifying thin clouds. However, on the CHLandsat8 dataset, the algorithm’s performance in handling boundary details was somewhat lacking, with a relatively high number of missed detected pixels, and was particularly prone to misclassifying similar background areas near cloud boundaries as clouds. As for the 95-Cloud dataset, our algorithm performed adequately in accurately locating thin cloud regions, but there were still minor flaws, specifically errors in pixel classification within some thin cloud boundary areas. This situation could be attributed to two main factors: firstly, the diversity in cloud shapes and background complexity across different datasets posed challenges for the algorithm in adapting to new environments; secondly, the generalization ability of the algorithm itself needs to be improved, especially in terms of feature extraction, where there are design deficiencies that make it difficult to fully capture and represent detailed features. Therefore, to further enhance the algorithm’s performance across different datasets, we need to conduct in-depth optimizations and improvements in these areas.

Based on the above analysis, in order to further enhance the performance of the algorithm across various datasets, especially in terms of improving its ability to handle boundary details and recognize complex cloud patterns, we plan to actively introduce and integrate advanced diffusion models into the algorithm framework in our subsequent research. These models have demonstrated remarkable potential in the fields of image processing and computer vision, particularly in capturing fine-grained features of images and enhancing the generalization performance of models. For instance, Rsdiff [50] is able to capture local and global features in images more efficiently by introducing a random wandering and diffusion process. Crs-diff [51] is a method based on cross-diffusion and adaptive feature extraction, which is capable of effectively removing noises and interfering with information while preserving the details of an image. Based on the introduction of the diffusion model, we will also carry out the in-depth optimization and adjustment of our algorithm. This will include fine-tuning the model parameters to adapt to the characteristics of different datasets, optimizing the feature selection strategy to improve the efficiency and accuracy of the algorithm, and introducing more diverse regularization methods to enhance the stability and generalization performance of the model. The aim will be to ensure that the algorithm can show excellent and stable performance on all types of datasets, especially in handling complex cloud morphology and fine boundary details.

## 4. Conclusions

With the advancement of remote sensing technology, it has become easier to acquire remote sensing images. The accuracy of cloud detection, as a preprocessing step to extract key information from images, is crucial to ensure the effective utilization of remotely sensed data. In this study, we propose a deep neural network based on a multi-feature fusion attention mechanism for cloud detection in remote sensing images by considering the advantages of codec structures and cloud image features for design. Firstly, we cleverly utilized dilated convolutions with varying dilation rates to capture multi-scale information, effectively enlarging the receptive field and enhancing the model’s ability to perceive global features. At the same time, by assigning differentiated weights to the channels, the model can focus more on important feature information. Secondly, to further improve model performance, we designed the SPFA module. This module ingeniously employs multi-scale strip convolution techniques to capture the long-distance spatial dependencies of cloud layers, thereby enabling a deeper understanding of their spatial distribution patterns. Additionally, the SPFA module integrates a self-attention mechanism to explore the intrinsic relationships between pixels. In addition, the DCA module is used to fuse low-level and high-level information in skip connections to improve the efficiency of semantic information propagation, enabling more feature details to be acquired while recovering the spatial resolution of the image. Our results from a series of experiments on the HRC_WHU, CHLandsat8, and 95-Cloud datasets showed a significant improvement in the cloud detection accuracy of our model compared to previous cloud detection methods. The visualization experiments demonstrated that our algorithm was able to generate smooth and accurate boundary predictions for complex cloud maps and also showed excellent localization ability for hard-to-see thin clouds. Nevertheless, we recognize that the algorithm may still lose some details when detecting extremely thin cloud regions. Therefore, we will continue to work on optimizing the algorithm to capture the details of debris and thin cloud features more accurately to further improve the comprehensiveness and accuracy of cloud detection. 

## Figures and Tables

**Figure 1 sensors-25-01245-f001:**
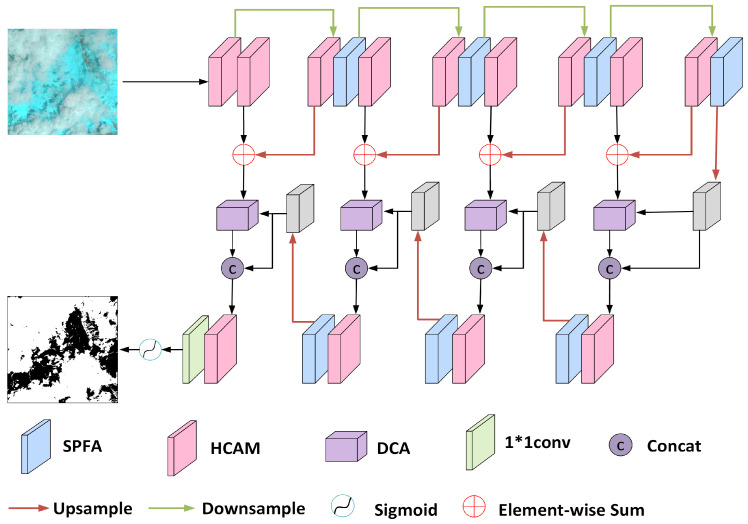
The structure of NFCNet.

**Figure 2 sensors-25-01245-f002:**
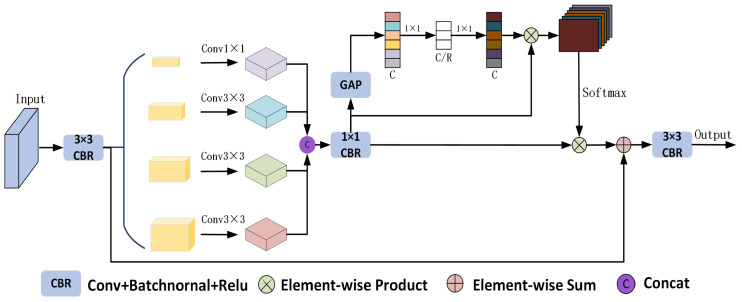
Diagram of HCAM structure.

**Figure 3 sensors-25-01245-f003:**
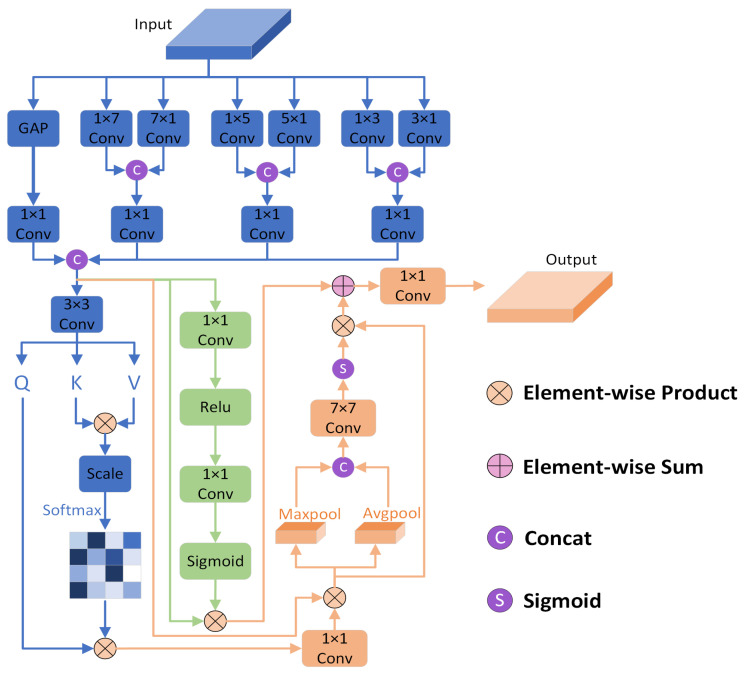
Diagram of SPFA structure.

**Figure 4 sensors-25-01245-f004:**
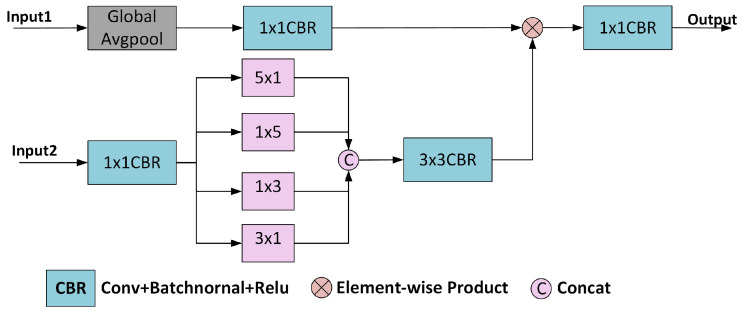
Diagram of DCA structure.

**Figure 5 sensors-25-01245-f005:**
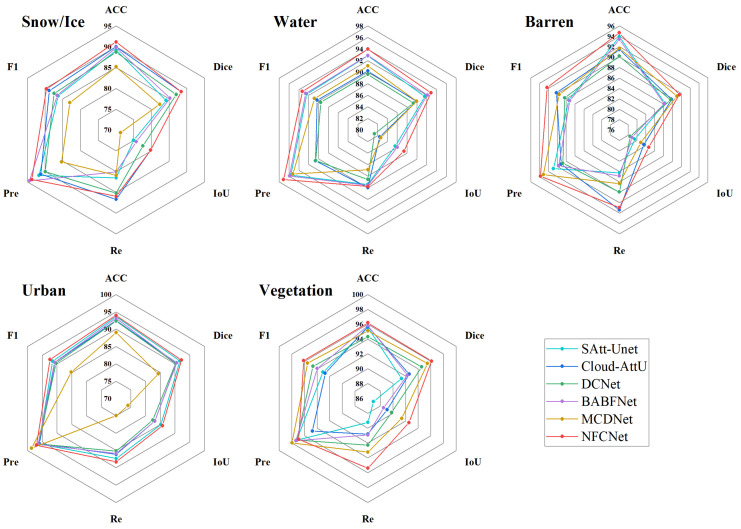
Comparison of accuracy of different methods with five land cover types.

**Figure 6 sensors-25-01245-f006:**
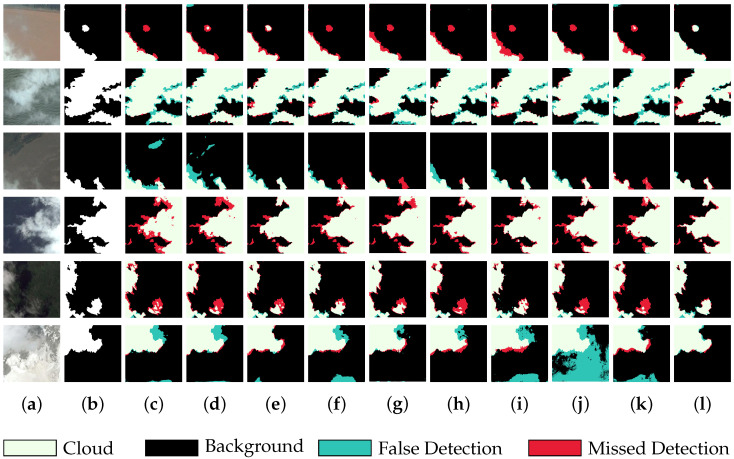
The visual output of each network on the HRC_WHU dataset: (**a**) the original image; (**b**) the corresponding label; (**c**) the prediction of UNet; (**d**) the prediction of deeplabv3+; (**e**) the prediction of SAtt-UNet; (**f**) the prediction of Cloud-Attu; (**g**) the prediction of DCNet; (**h**) the prediction of CRSNet; (**i**) the prediction of BABFNet; (**j**) the prediction of MCDNet; (**k**) the prediction of AFMUNet; and (**l**) the prediction of NFCNet.

**Figure 7 sensors-25-01245-f007:**
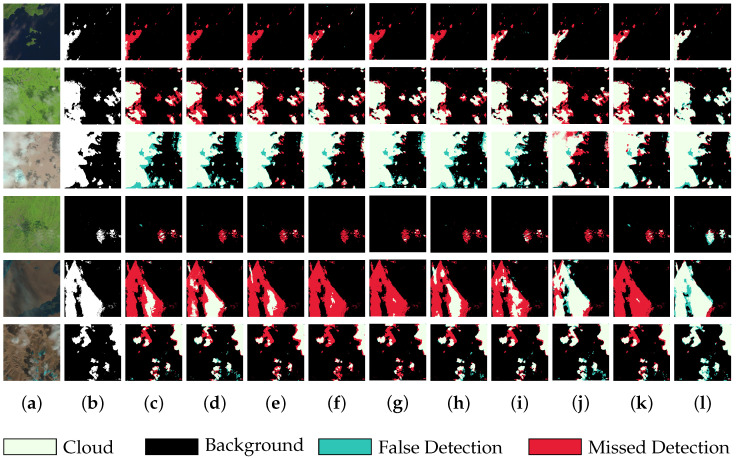
The visual output of each network on the CHLandsat8 dataset: (**a**) the original image; (**b**) the corresponding label; (**c**) the prediction of UNet; (**d**) the prediction of deeplabv3+; (**e**) the prediction of SAtt-UNet; (**f**) the prediction of Cloud-Attu; (**g**) the prediction of DCNet; (**h**) the prediction of CRSNet; (**i**) the prediction of BABFNet; (**j**) the prediction of MCDNet; (**k**) the prediction of AFMUNet; and (**l**) the prediction of NFCNet.

**Figure 8 sensors-25-01245-f008:**
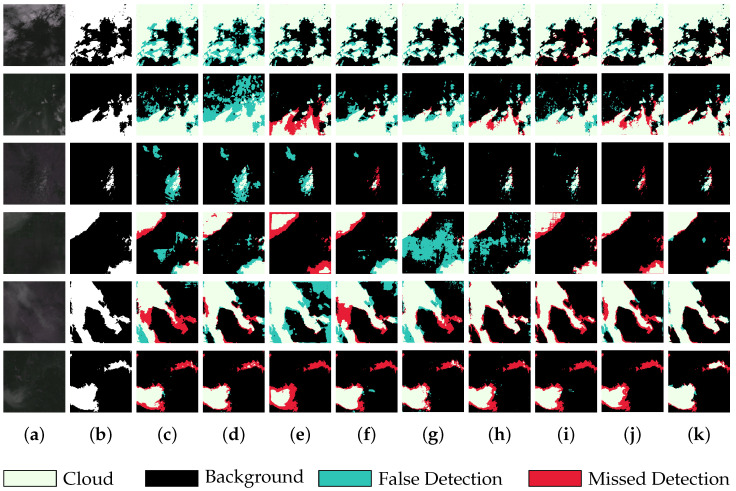
The visual output of each network on the 95-Cloud dataset: (**a**) the original image; (**b**) the corresponding label; (**c**) the prediction of UNet; (**d**) the prediction of deeplabv3+; (**e**) the prediction of SAtt-UNet; (**f**) the prediction of Cloud-Attu; (**g**) the prediction of DCNet; (**h**) the prediction of CRSNet; (**i**) the prediction of BABFNet; (**j**) the prediction of AFMUNet; and (**k**) the prediction of NFCNet.

**Table 1 sensors-25-01245-t001:** Quantitative comparison of different methods on HRC_WHU dataset (IN %).

Method	Acc	Dice	mIoU	Re	Pre	F1	Param Size (M)	FLOPs (G)
Baseline	92.94	91.00	83.87	90.01	92.40	91.19	3.4692	4.1324
HCAM	94.36	92.97	87.11	92.95	93.28	93.11	5.6700	8.4720
HCAM + SPFA	95.16	94.21	89.15	94.21	95.15	94.67	14.7034	18.3244
HCAM + SPFA + DCA	95.20	94.58	89.85	94.15	94.80	94.47	14.7759	18.5641
HCAM + SPFA + HCAM + DCA (ours)	95.44	94.86	90.25	94.03	95.74	94.88	16.0874	22.5490

**Table 2 sensors-25-01245-t002:** Quantitative comparison of different methods on HRC_WHU dataset (IN %).

Method	Acc	Dice	mIoU	Re	Pre	F1
UNet	92.94	91.00	83.87	90.01	92.40	91.19
DeeplabV3+	93.62	92.35	86.29	94.05	91.20	92.60
SAtt-UNet	94.95	94.08	88.97	93.88	94.45	94.16
Cloud-AttU	93.86	92.79	86.77	93.31	92.54	92.92
DCNet	93.81	92.63	86.49	92.86	92.63	92.74
CRSNet	93.12	91.74	85.07	92.30	91.59	91.95
BABFNet	94.31	93.25	87.56	93.00	93.68	93.36
MCDNet	94.16	93.36	87.60	93.21	93.61	93.41
AFMUNet	93.11	91.39	84.47	91.05	92.11	91.58
NFCNet (ours)	95.44	94.86	90.25	94.03	95.74	94.88

**Table 3 sensors-25-01245-t003:** Quantitative comparison of different methods on CHLandsat8 dataset (IN %).

Method	Acc	Dice	mIoU	Re	Pre	F1
UNet	91.21	86.10	76.86	92.98	81.52	86.87
DeeplabV3+	91.75	86.13	77.76	88.23	85.76	86.97
SAtt-UNet	91.90	86.80	78.01	90.91	84.54	87.60
Cloud-AttU	92.58	86.83	78.08	85.29	90.13	87.64
DCNet	91.86	86.49	77.47	89.33	85.29	87.26
CRSNet	90.40	85.40	75.88	94.08	79.68	86.28
BABFNet	92.55	87.04	79.11	90.89	86.19	88.48
MCDNet	92.49	86.72	77.84	84.68	90.35	87.42
AFMUNet	92.17	86.89	78.14	88.44	86.95	87.68
NFCNet (ours)	92.96	87.89	80.61	91.94	87.88	89.02

**Table 4 sensors-25-01245-t004:** Quantitative comparison of different methods on 95-Cloud dataset (IN %).

Method	Acc	Dice	mIoU	Re	Pre	F1
UNet	92.45	84.10	75.53	86.85	84.36	85.58
DeeplabV3+	92.43	84.95	76.55	92.72	80.98	86.45
SAtt-UNet	93.15	84.51	76.38	85.95	86.35	86.14
Cloud-AttU	94.14	85.82	78.02	86.83	87.07	86.94
DCNet	92.73	84.59	76.11	89.33	82.86	85.97
CRSNet	91.03	82.19	73.01	88.91	79.46	83.91
BABFNet	94.94	88.15	81.37	89.01	88.88	88.94
AFMUNet	93.93	86.29	78.99	88.17	86.59	87.37
NFCNet (ours)	95.45	89.53	83.20	90.10	90.64	90.36

## Data Availability

The data provided in this study are available from the first author on request.

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
