# Peer review of "Cloud Detection in Remote Sensing Images Based on a Novel Adaptive Feature Aggregation Method"

_sensors, 2025, doi:10.3390/s25041245_

Round 1

Reviewer 1 Report

Comments and Suggestions for Authors

Please refer to the attachment

Comments on the Quality of English Language

Please refer to the attachment

Reviewer 2 Report

Comments and Suggestions for Authors

The paper "Cloud Detection in Remote Sensing Images Based on a Novel Adaptive Feature Aggregation Method" presents a novel and highly relevant approach to the field of cloud detection in remote sensing images. Written by Wanting Zhou, Yan Mo, Qiaofeng Ou, and Shaowei Bai, the paper addresses persistent challenges such as the accurate identification of cloud boundaries and thin clouds in complex scenarios, and proposes the NFCNet model to address these difficulties.

The presented model integrates three main modules: the Hybrid Convolutional Attention Module (HCAM), which captures multi-scale information and assigns different weights to the most important channels; the Spatial Pyramid Fusion Attention Module (SPFA), which compensates for detail losses during the downsampling process and improves segmentation between clouds and non-clouds; and the Dual-Stream Convolutional Aggregation Module (DCA), which combines information from different levels to improve detail sensitivity and segmentation.

The experiments performed demonstrated superior results compared to existing methods, using robust datasets such as HRC_WHU, CHLandsat8, and 95-Cloud. The proposed model stood out for its ability to finely segment cloud contours and identify thin clouds, even in highly complex scenarios, in addition to presenting high metrics in terms of precision (PA), Dice, IoU, and F1. Therefore, I recommend this paper.

Reviewer 3 Report

Comments and Suggestions for Authors

The authors proposed a deep learning network model, called NFCNet, for cloud detection in remote sensing images, a relevant topic for several applications, such as meteorology, agriculture, and environmental monitoring. Comparative experiments were conducted with other models such as Unet, DeepLabV3+, and SAtt-Unet, showing that the proposed model outperforms the previous ones in several aspects. The work is well written, well organized, and the research gaps and contributions were presented. My suggestions for improving the quality of the manuscript are:

  1. The paper does not present information about the processing time and computational requirements of the proposed model. Considering the high computational cost of deep networks, it would be relevant to include an analysis of inference time and hardware use.
  2. The paper shows that the model works well on the datasets used, but does not discuss its applicability in other scenarios (e.g., different atmospheric conditions, sensor types, varied resolutions).
  3. The experimental results section could include a subsection to discuss the results.
